# In Situ Ternary Boride: Effects on Densification Process and Mechanical Properties of WC-Co Composite Coating

**DOI:** 10.3390/ma13081995

**Published:** 2020-04-24

**Authors:** Junfeng Bao, Yueguang Yu, Bowen Liu, Chengchang Jia, Chao Wu

**Affiliations:** 1Institute for Advanced Materials and Technology, University of Science and Technology Beijing, Beijing 100083, China; Bao_jf@bgrimm.com (J.B.); lbw_1211@126.com (B.L.); jcc@ustb.edu.cn (C.J.); 2BGRIMM Technology Group, Beijing 100160, China; 18101380412@163.com; 3Beijing Engineering Technology Research Center of Surface Strengthening and Repairing of Industry Parts, Beijing 102206, China

**Keywords:** WC–Co, Mo–B_4_C, in situ reaction, ternary boride, densification process

## Abstract

New coatings resistant to corrosion in high-temperature molten zinc aluminum were prepared by supersonic flame spraying of various composite powders. These composite powders were prepared by mixing, granulation, and heat treatment of various proportions of Mo–B_4_C powder and WC and Co powder. X-ray diffraction (XRD), transmission electron microscopy (TEM), scanning electron microscopy (SEM), high-angle annular dark-field scanning transmission electron microscopy (HAADF–STEM), energy dispersive X-ray spectroscopy (EDS), and mechanical analysis were used to study the effects of Mo–B_4_C on the microstructure, phase, porosity, bonding strength, and elastic modulus of the composite powder and coating. Results show that the addition of an appropriate quantity of Mo–B_4_C reacts with Co to form ternary borides CoMo_2_B_2_ and CoMoB. Ternary boride forms a perfect continuous interface, improving the mechanical properties and corrosion resistance property of the coating. When the amount of Mo–B_4_C added was 35.2%, the mechanical properties of the prepared coating reached optimal values: minimum porosity of 0.31 ± 0.15%, coating bonding strength of 77.81 ± 1.77 MPa, nanoindentation hardness of 20.12 ± 1.85 GPa, Young’s modulus of 281.52 ± 30.22 GPa, and fracture toughness of 6.38 ± 0.45 MPa·m^1/2^.

## 1. Introduction

Hot-dip galvanizing is a very economical and efficient method to prevent steel corrosion during atmospheric exposure [1]. However, the wear and corrosion of sink rolls and other components during the galvanizing process are major challenges in the galvanizing industry. Each year, the loss caused by component corrosion across the ~57 hot-dip galvanizing production lines in the United States is nearly 500 million US dollars [2,3,4]. Moreover, the surface state of the sink rolls directly affects the quality of the galvanized steel surface; hence, avoiding sink roll corrosion is important. The corrosion rate of common alloy and steel products is faster because of the wettability of the metal in molten zinc aluminum. W, Mo, B, side elements, and other elements are not wettable in molten zinc; thus, they can be used as corrosion-resistant sink rolls in hot-dip galvanizing. On the one hand, the plasticity and impact resistance of W–Mo alloys are poor. On the other hand, W–Mo alloys are too expensive for use as sink rolls, limiting their use [5,6]. Therefore, typically, a protective coating is applied to prevent the sink rolls from corroding in molten zinc. In recent years, WC–Co, Al_2_O_3_, Zr_2_O_3_–Y_2_O_3_, MoB–CoCr, and other coatings have been used in sink roll anti-corrosion coatings. According to the different properties of the coatings, they can be roughly divided into cermet coatings, oxide ceramic coatings, and new MoB–CoCr coatings. Of these, the cermet coatings, represented by WC–Co, exhibit better corrosion resistance and wear resistance. Simultaneously, owing to the existence of metal binder, the coating has better toughness under the premise of maintaining high hardness [7,8,9,10]. Its main anti-corrosion principle is to form an aluminum-rich phase as a protective film to isolate the molten zinc aluminum. However, owing to the high Co metal content, the coating easily yields to metal ternary corrosion, leading to coating damage [11]. Al_2_O_3_ and Zr_2_O_3_–Y_2_O_3_ have high melting points, making them almost impervious to molten zinc aluminum; they are primarily used as oxide ceramics to prepare anticorrosive coatings via plasma spraying. However, the disadvantage of oxide ceramic coatings is poor toughness [12]. In recent years, a new type of thermal spraying material, MoB–CoCr, has become widely used in molten zinc aluminum corrosion-resistant hot-dip galvanizing [13,14,15]. The service life of a MoB–CoCr coating is more than 600 h.

Because of the high economic impact of solutions to these problems, scholars have remained committed to the study of the surface coatings of sink rolls used for preparing hot-dip galvanized steels. However, many scholars have studied the characteristics of MoB–CoCr coatings. The durability of MoB–CoCr coatings in the hot-dip galvanizing industry is clearly better than those of other alloys owing to the non-wettability of boride in molten zinc, the compactness of the coating with its consequently lower defect rate, and the superior mechanical properties, all meeting the needs of sink roll use. Boride has gradually substituted WC cermet coatings because of the poor wettability of molten zinc–aluminum, particularly in molten zinc–aluminum with high aluminum content at temperatures up to 600 °C [16,17].

Boride technology development can be traced through a series of key studies. Sugiyama et al. [18] successfully prepared a WC–WB–W_2_B composite coating via in-situ synthesis based on B_4_C–W–WC. Their coating exhibited high density, excellent mechanical properties, and high resistance to molten zinc corrosion because of the high B content of its ceramic phase. In the past two decades, WC–Co coatings have been widely used as sink roll surface protection materials. The corrosion-resistance of these coatings remains mainly limited by their high Co metal content. Wang et al. [19] proposed the use of Co_6_W_6_C and Co_3_W_3_C phases instead of Co to improve the corrosion resistance of WC–Co. However, this proposal produced an excess of brittle phases, thereby reducing the toughness of the WC–Co coating. Therefore, nano WC powders were instead used for preparation, and good corrosion resistance was achieved. These methods have their own advantages, but the use of nano WC powder increases the cost of the coating for all such methods. In contrast, too high a proportion of the ceramic phase risks losing the toughness of the coating, resulting in a decline in mechanical properties. In the actual production process, boride ceramic coating is commonly used as a corrosion-resistant coating material at 700 °C, under which the steel softens because of the high temperature such that the boride ceramic coating can meet the conditions. The modified WC–Co coating is usually used at 630 °C to achieve higher toughness.

Herein, a WC–Co coating modified using Mo–B_4_C was used to prepare corrosion-resistant coatings under working conditions of about 600 °C. The in-situ reaction of Mo–B_4_C and Co to form a ternary boride binder was investigated, as were the effects of the in-situ reaction on the coating’s porosity, interface, and mechanical properties. A low-porosity coating rich in CoMoB and CoMo_2_B_2_ ternary boride, which could enhance resistance to molten zinc–aluminum corrosion, was successfully prepared with mechanical properties equivalent to those of traditional WC–Co coatings, but stronger resistance to molten zinc–aluminum corrosion due to the ternary boride.

## 2. Materials and Methods

### 2.1. Preparation of Composite Powder

In this experiment, WC powder (average particle size 1.10 μm; Xiamen Jinlu Special Alloy Co., Ltd., Xiamen, China), Mo powder (average particle size 1.75 μm; BGRIMM Advanced Material Science & Technology Co., Ltd., Beijing, China), Co powder (average particle size 1.52 μm; Zhejiang Huayou Cobalt Industry Co., Ltd., Jiaxing, China), and B_4_C powder (average particle size 1.95 μm; Nanuo Advanced Materials Co., Ltd., Guangzhou, China) were selected as raw material powders. Mixing, granulation, and heat treatment were applied to prepare the composite powder.

Each of the four powders was mixed according to the powder mass ratio shown in Table 1. A planetary ball mill was used for full blending, utilizing 5 μm WC grinding balls at a ball:material ratio of 5:2. A 3 wt.% polyethylene glycol (PEG4000) aqueous solution in deionized water was added as a ball mill additive. The speed was maintained at 80 rad/min for 24 h.

An LGZ-50 centrifugal spray dryer was used for granulation of the mixed powder slurry. The inlet temperature was 300 °C, the outlet temperature was 120 °C, and the slurry flow rate was 650 mL/min. Binder removal and heat treatment were performed in a tubular resistance furnace. To avoid other reactions, an argon gas shielding atmosphere was selected for the heat treatment, with a gas flow rate of 1 L/min. The furnace temperature was increased at a rate of 10 °C/min to 180 °C, then 1 °C/min to 420 °C, then maintained at 420 °C for 2 h to ensure PEG binder removal. To ensure that the heating process would not crack the sample under excessive internal stress, the temperature was increased to 600 °C at which it was then maintained for 1 h. In the powder reaction stage, the temperature was increased at a rate of 10 °C/min to 1000 °C heat preservation for 0.5 h, then increased at a rate of 5 °C/min to 1260 °C heat preservation and maintained at 1260 °C for 3 h to ensure full reaction, followed by furnace cooling. After heat treatment, the powder was ground and sieved with 325 mesh and 800 mesh sieves to obtain a composite powder with a particle size distribution of 18–45 nm.

### 2.2. Preparation of Coatings

The composite powder was used for supersonic flame spraying, with 316L stainless steel used as a matrix. Considering the large difference in thermal expansion rates between the two materials, NiCrAlY alloy was chosen as the intermediate bond layer. The coating was prepared using a JP-8000 supersonic flame sprayer (PROXAIR company, Boston, MA, USA). After cleaning and sandblasting the 316L stainless steel matrix, the NiCrAlY intermediate layer, and the composite coating were successively sprayed. The spraying distance was 360 mm, and the coating thickness was about 250 μm. The microstructure and mechanical properties of the prepared coating were then further analyzed.

### 2.3. Analysis

The effects on the microstructure, morphology, densification, and mechanical properties of the composite powder and coating of adding Mo–B_4_C were analyzed. A SU 5000 SEM (HITACHI, Tokyo, Japan) was used to observe the microstructure of the raw material, composite powder, and coating. A D8 ADVANCE X-ray diffractometer (Cu K_α_, λ = 0.154184 nm; BRUKER, Karlsruhe, Germany), was used to analyze the phase contained in the composite powder at a scanning speed of 10°/min. The microstructure of the coating was analyzed using a JEOL JEM-1010 TEM, whereas the microscopic phase of the coating was analyzed via selected area electron diffraction (SAED). The diffusion forms of elements in each phase were determined using mapping in scanning transmission electron microscopy (STEM) mode. The samples observed using transmission electron microscopy were cut using a focused ion beam (FIB).

The epoxy adhesive coating was selected in accordance with ASTM C633-79, and the bonding strength of the coating was tested using a WDW-100A electronic universal testing machine. The porosity of the coating was measured in accordance with ASTM E2109-01. Image J software was used to calculate the porosities of 30 coating tissue photos for each sample. CSM-NHT2 nanometer indentation measurements of loading–unloading were used to measure the elastic modulus of the coating cross section and the nanometer hardness. Testing conditions were as follows: a Berkovich pressure head was used for linear loading, the maximum load was 10 mN, the loading and unloading rates were both 20 mN/min, and the pressure was maintained for 5 s under the maximum load. The fracture toughness (*K_IC_*) of the coating was characterized using the Vickers indentation method, as measured using a 402MVATM Vickers hardness tester. The test parameters were as follows: loading load was 300 g, holding pressure was 10 s. Fracture toughness (*K_IC_*) was evaluated according to the half-diagonal and crack length of the indentation after each indentation. The fracture toughness of the coating was calculated as
(1)$$K_{IC}=0.079\frac{P}{a^{3/2}}\log\left(\frac{4.5a}{c}\right),$$
where *P* is the applied load in N, *a* is the length in micrometers of the half-diagonal of the indentation, and *c* is the length in micrometers from the center of the indentation to the end of the crack. The formula applied was 0.6 ≤ *c*/*a* ≤ 4.5, and for each sample at least 10 points were measured. These samples with different coating were tested for molten zinc aluminum corrosion. The experimental equipment was a RJ-75-6 crucible furnace. The composition of the molten zinc aluminum was Al-43.5%Zn-1.5%Si, the experimental temperature was controlled at 630 °C for 72 h. Then, the microstructure of the coating after 72 h corrosion of different coatings was analyzed using SEM and EDS.

## 3. Results and Discussion

### 3.1. Characterization of the Raw Materials

The surface morphologies are shown in Figure 1 for the four raw material powders B_4_C, Co, Mo, and WC. The B_4_C particle morphology selected in the experiment was that of an irregular block with edges and corners. Co particles were subspherical or dumbbell-shaped, agglomerating together. Mo particles were larger spherical or subspherical particles, also showing agglomeration. WC particle size was relatively small, with a more irregular shape.

### 3.2. Characterization of the Prepared Composite Powders

Powders with different ratios of raw materials were prepared through the process of mixing, granulation, and heat treatment to form composite powders. These were fully mixed to form metallurgical bonds; chemical reactions then produced specific phase compositions. Phase analyses of these composite powders are shown in Figure 2a,b. Sequentially, from sample #6 through sample #1, the quantity of Mo–B_4_C added increased from 0 wt. % to 88 wt. %, dictating the range of phases of the mixed powders post heat treatment. Sample #6 was a typical WC–12Co cemented carbide powder. Post granulation and heat treatment, WC and Co failed to react to form a new phase; hence, the WC and Co physical phases were retained. The phase compositions of samples #3 through #5 were similar. With increasing amount of added Mo–B_4_C, the relative peak strength of Co gradually decreased, whereas the relative peak strength of ternary-boride CoMoB and CoMo_2_B_2_ generated by the reaction of Co with Mo–B_4_C gradually increased. This pattern indicates that Co and Mo–B_4_C consumed the original Co and produced two ternary borides, CoMoB and CoMo_2_B_2_, and that the quantity of these products gradually increased with increasing Mo–B_4_C addition. Literature survey and XRD results indicate that the reactions occurring during the heat treatment were those shown in Equations (2) and (3) [20].
5Mo + B_4_C → 4MoB + MoC(2)
2Co + 3MoB → CoMo2B2 + CoMoB(3)

As Mo participated in the reaction to generate MoC, some of the MoC was decarburized at high temperature during the heat treatment such that (Mo,W)_2_C was observed in the phase of sample #3. With increasing addition of Mo–B_4_C, the MoB_2_ phase was observed in samples #1 and #2, and the relative peak strength of MoB_2_ increased with increasing addition of Mo–B_4_C. This behavior can be explained by noting that the amount of Co added was fixed at 12%, whereas the additive amount of Mo–B_4_C was excessive such that part of the Mo reacted with B_4_C to generate MoB_2_. Notably, the chemical reactions describing the full process of preparing the composite powder were relatively complicated; the reactions in Equations (2) and (3) [20] only represent the main reactions. The presence of free Co was observed even in sample #1, of which Mo–B_4_C constituted 88 wt. %.

Figure 3 shows the surface morphology and cross-sectional morphology after heat treatment of composite powders comprising different ratios of the same raw materials. The surface morphology shows that owing to the large amount of Mo–B_4_C addition, most of the bonded phase Co and Mo–B_4_C reacted to generate ternary boride CoMoB and CoMo_2_B_2_. Moreover, because of excessive addition of Mo–B_4_C, MoB or MoB_2_ was generated. These reactions thus resulted in a large reduction of bonded-phase Co and generated boride ceramic phases.

This outcome is more obvious in Figure 3b1. Owing to the small amount of the bonded phase, more pores are visible on the surface of the composite powder, and the particles are fine and broken. The surface morphologies of samples #3, #4, and #5 are similar. Co reacts with Mo–B_4_C to produce a ternary boride ceramic phase binder. Additionally, some of the Co is retained as the bonding phase, with fewer surface defects. Simultaneously, the raw material powder is well bonded, and the particle sizes are evenly mixed to form a composite powder. Furthermore, samples #3–#6 show WC powder particles of smaller size adhered to the composite powder. Sample #6 shows the typical WC–12Co granulation morphology: the Co bonding phase is fully enveloped by the smaller WC hard phase particles, with no pore defects visible on the surface. The cross-sectional morphologies of samples #2, #4, and #6 are shown in Figure 3b2–f2. The internal binding of #2 composite powder particles with excess Mo–B_4_C was poor, and no metallurgical binding formed between the particles. However, for #4 composite powder the intergranular combination was good, and the powder combination was good. Although some pores remain visible, no obvious interface is evident. The interior of the #6 powder composite was a combination of WC and Co. WC and Co show good macroscopic combination, with few defects. The addition of Mo–B_4_C and Co resulted in an in-situ reaction, and the ternary boride thus formed intersected with WC. The binding state of each raw material powder in the composite powder further affects the microstructure and mechanical properties of the prepared coating.

### 3.3. Effect of Mo–B_4_C on the Morphology and Densification Process of the Coating

The composite coatings with different powder composition ratios were prepared using supersonic flame spraying. Typical #2, #4, and #6 coating tissues were selected for comparison, respectively, representing the excess addition of Mo–B_4_C coating (#2), suitable addition of Mo–B_4_C coating (#4), and absence of Mo–B_4_C (#6). The cross-sectional microstructures of the coatings are shown in Figure 4b3–f3, showing that coatings with different ratios combine well with the matrix layer. The microstructure of coating #2 is shown in Figure 4b3. Owing to the lack of a cohesive Co phase and the excessive addition of Mo–B_4_C, a large quantity of MoB or MoB_2_ was generated, resulting in numerous pore defects in the coating microstructure. Large pores remained in the coating, with diameters around 10 μm. Particles inside the pores failed to combine with the matrix, and microcracks were generated around them. Owing to the existence of pores, some particles were not fully bonded, and micro-cracks seriously diminished the coating strength and mechanical properties.

The structure of coating #4 is shown in Figure 4d3. The coating structure is uniform, and no pores can be observed. The enlarged diagram shows that the numerous hard phases generated are evenly distributed in the matrix. The microstructure of coating #6 is shown in Figure 4f3. This coating, lacking Mo–B_4_C, also shows certain pore defects. The enlarged structure shows that WC particles are evenly distributed in the matrix of the Co bonding phase; however, the combination of Co and WC is not good. Thus, tiny pores exist at the interfaces between the WC particles and Co.

The coating’s microstructure and defects both affect the bonding strength of the coating. Therefore, we measured the porosity of coatings with different compositional ratios and calculated their bonding strengths, as shown in Figure 4g. With increasing addition of Mo–B_4_C, the porosity of the coating first decreased and then increased, with the #4 coating exhibiting a porosity minimum. In coatings #6, #5, and #4, the addition of Mo–B_4_C reacted with Co to produce ternary boride: CoMoB and CoMo_2_B_2_. This in-situ reaction resulted in better interfacial bonding between Co and reaction products. Simultaneously, Mo and W showed better wettability and mutual diffusion ability, resulting in a better interface between Mo and W. Therefore, Mo and B_4_C can act as the media to form a perfect interface between Co and WC.

In coatings #4, #3, #2, and #1, the content of Mo and B_4_C, increased gradually. The bonded-phase Co was largely consumed, making it difficult for the excess ceramic phase in the matrix to be completely densified. In contrast, when Mo and B_4_C were in excess, MoB or MoB_2_ was generated. This binary boride was the same as WC, and the difficulty it showed in combining with the matrix resulted in further porosity increase. The porosity of a given coating affects its bonding strength. Among the coatings with Mo and B_4_C, coating #4 showed the highest bonding strength, owing to the reduction in defects. Compared with coating #4, coating #6 without Mo and B_4_C showed a slightly higher bond strength owing to the higher bond strength of metal Co compared with ternary boride as the bond phase. Even though the porosity was high, the numerous metal bond phases provided a higher bond strength.

### 3.4. Microstructural Evolution of Coating

To further analyze the changes in coating microstructure, bonding phase, and hard phase during supersonic flame spraying, TEM was used in combination with selective electron diffraction and EDS to analyze coating microstructures at different compositional ratios. Figure 5a shows the clear field image of the microstructure of coating #2. Because of the high temperature of supersonic flame spraying, the matrix is amorphous. The hard phase is evenly distributed in the matrix. The excess Mo and B_4_C in sample #2 reveal WC and MoB particles in the coating wherein WC particles appear as irregular angular blocks, whereas MoB particles are spherical. Meanwhile, in the images of coating #4 in Figure 5b, the coating matrix is also amorphous. The ternary borides CoMoB and CoMo_2_B_2_ are uniformly distributed in the matrix.

The locations of W, Mo, and Co and the interface conditions were further analyzed in sample #4, which showed the lowest porosity and the best bonding strength. Figure 6a shows a TEM image of the microstructure of coating #4. The particle has internal and external structure, and its overall diameter is 34.28 nm. The energy spectrum of the inner structure shows only W and C, indicating only WC molecules, whereas the outer structure contains W, Mo, and C. Figure 6b, showing the selected electron diffraction spot at the interface of the inner and outer layers of the particle, includes the diffraction spots of two phases, namely WC and CoMo_2_B_2_, under the same belt axis. According to the energy spectrum data, the inner structure should be WC, whereas the outer structure should be CoMo_2_B_2_. Moreover, the two phases of CoMo_2_B_2_ and WC show a certain crystallographic orientation relationship, indicating that CoMo_2_B_2_ and WC have a good interface combination, ideally a perfect interface. Sun et al. [21], found that the in-situ reaction of B_4_C led to forming a continuous perfect interface. Compared with discontinuous interface, perfect interface can avoid crack initiation at the interface between hard particle and matrix. Therefore, the action of the bearing force of a hard particle will not form microcracks in this system, as the transmission of stress will not become discontinuous. Figure 6c is the high-angle annular dark-field (HAADF)-STEM mode mapping diagram, showing (1) an obvious mutual diffusion between the Mo in the particle exterior and W in the particle interior, and the (2) perfect combination of W and Co. This finding further demonstrates the continuity of the perfect interface. In sum, the ternary borides CoMo_2_B_2_ and CoMoB, formed with appropriate addition of Mo and B_4_C in sample #4, can be used as the medium to form a perfect interface between WC and Co. In the presence of the boride ceramic bonding phase, the internal pores of the coating are reduced, and the particles are well combined with the matrix. The existence of a perfect interface further optimizes the mechanical properties of the coating. Simultaneously, ternary boride is used as the ceramic bonding phase instead of Co metal, reducing the number of defects in the coating. This defect reduction can effectively enhance the corrosion-resistance of the coating in high-temperature molten zinc aluminum.

### 3.5. Mechanical Property Evolution of Coating by Mo–B_4_C Addition

Figure 7a shows the changes in nanoindentation hardness and Young’s modulus with the compositional trend among coatings. As the amount of Mo + B_4_C increased, the nanoindentation hardness tended to first increase and then decrease. This pattern can be explained as follows: the added Mo + B_4_C reacted with Co to form a hard phase; hence, the nanohardness increased initially. However, when the added amount of Mo + B_4_C exceeded 35.2 wt. %, Mo + B_4_C was in excess, generating binary borides instead of ternary borides. Although the hardness of MoB is higher than that of CoMo_2_B_2_, the appearance of MoB was accompanied by the formation of pore defects and a poor combination between MoB and matrix. These behaviors resulted in a decline in nanohardness as the amount of Mo + B_4_C continued to increase. A similar pattern was observed in Young’s modulus. The only difference was that the Young’s modulus of the coating without Mo + B_4_C was slightly higher than that of the optimal additive coating #4. This disparity can be explained as follows: the coating structure without Mo + B_4_C was mainly Co matrix combined with WC hard phase, and numerous Co metal bonding phases provided a high Young’s modulus. Figure 7b shows the change in fracture toughness of the coating with increasing addition of Mo + B_4_C; the trend in fracture toughness is similar to that of Young’s modulus. In contrast with sample #6, which lacked Mo + B_4_C, the other samples’ reactions consumed part of the Co metal bonding phase. With this change in the ceramic phase, the fracture toughness of the coating was reduced to a certain extent. As the addition of Mo + B_4_C continued to increase to 35.2 wt. %, numerous ternary borides were generated to replace Co as the ceramic bonding phase, and the interfacial bonding was better than that of Co–WC. This phenomenon promoted the coating’s increase in fracture toughness. As the addition of Mo + B_4_C continued to increase, the formation of a large number of defects and the formation of binary borides such as MoB lost the advantage of good interfacial bonding between ternary borides and Co–WC, which resulted in the coating’s decrease in fracture toughness.

### 3.6. Corrosion Resistance Property Evolution of Coating by Mo–B_4_C Addition

Figure 8 shows the cross-section structure of the composite coating with different Mo–B_4_C addition after 72 h and 630 °C molten zinc aluminum corrosion. It can be seen that most of the coating and NiCrAlY intermediate layer are corroded, and the surface structure is zinc aluminum alloy, as shown in Figure 8a 1# coating sample. This is due to the poor bonding ability of the structure and the existence of a large number of pore defects in the 1# coating, which makes it easier for the molten zinc aluminum to etch the coating and make the coating peel off. Figure 8b–e shows the difference of the thickness of the corrosion layer of the composite coating with different Mo–B_4_C addition. After 72 h of molten zinc aluminum corrosion testing, the corrosion layer of 4# coating is the thinnest, with an average thickness of 21.37 μm, showing the best corrosion resistance performance of molten zinc aluminum. On the one hand, the better interface combination makes it difficult for the molten zinc aluminum corrosion to diffuse inward through defects; on the other hand, the good corrosion resistance of the ternary boride also makes the corrosion resistance of the 4# coating greatly improved. However, for the 6# 12 Co-WC coating without Mo–B_4_C addition, the whole coating is corroded and there are a lot of pores in the coating due to the reaction of Co as a metal bonding phase with molten zinc aluminum.

In conclusion, the microstructure of WC–12Co changed with the addition of Mo + B_4_C and showed a trend with increasing addition of Mo + B_4_C. The microstructural change caused corresponding changes in the mechanical properties and corrosion resistance property of the coating. These changes can be assigned to two processes: (1) Mo + B_4_C was added to WC–12Co to form ternary borides CoMo_2_B_2_ and CoMoB. Because Mo and W showed good mutual diffusion and because Co could participate in the reaction to form ternary borides in situ, ternary borides CoMo_2_B_2_ and CoMoB showed perfect interfaces with WC and Co. This interfacial perfection increased the density, hardness, and fracture toughness of the coating, which all reached maximum values at 35.2 wt. % Mo + B_4_C. In this process, the addition of Mo + B_4_C served as the medium for a perfect interface between WC and Co. Simultaneously, the consumption of a large amount of Co and the generation of a ceramic bonding phase provided conditions that improved the corrosion resistance of the coating to molten zinc aluminum. (2) With the excess addition of Mo + B_4_C, the excessive consumption of the Co metal bonding phase reduced the coating density. Simultaneously, the excess Mo + B_4_C resulted in the generation of binary boride MoB, which contributed to the formation of numerous defects and imperfect interfaces. These interfaces and defects contributed to the deterioration of mechanical properties. Therefore, the optimal addition amount of Mo + B_4_C is 35.2 wt. %: 31.5 wt. % Mo, 3.7 wt. % B_4_C. At this composition, the mechanical properties were optimal: the minimum porosity was 0.31 ± 0.15%, the coating bonding strength reached 77.81 ± 1.77 MPa, the nanoindentation hardness was 20.12 ± 1.85 GPa, the Young’s modulus was 281.52 ± 30.22 GPa, and the fracture toughness was 6.38 ± 0.45 MPa·m^1/2^. The mechanical properties of the in-situ ternary boride composite coating are comparable to that of WC Co, and the ternary boride can effectively prevent the coating from being corroded by high temperature molten zinc. The corrosion-resistant wc-wb-w2b ceramics prepared by Sugiyama et al. [18] have an initial fracture toughness of less than 5.8 MPa·m^1/2^ and a Young’s modulus of up to 700 GPa. In contrast, the ternary boride composite coating has the advantages of mechanical properties, and the difference of Young’s modulus between the ternary boride composite coating and the matrix is smaller. Compared with ceramic coating, the composite coating prepared has better toughness and adaptability.

## 4. Conclusions

Composite powders of different proportions of Mo–B_4_C, WC, and Co were prepared by mixing, granulation, and heat treatment. When a small amount of Mo–B_4_C was added, it reacted with Co to produce ternary borides CoMo_2_B_2_ and CoMoB. When Mo–B_4_C was added to excess, Mo and B_4_C further reacted to generate MoB and MoB_2_.Boride ceramics were formed in situ to replace part of the Co bonding phase to improve corrosion resistance. When the amount of Mo–B_4_C added was 35.2%, some pores were present in the composite powder, but the combination between components was good. When Mo–B_4_C was added to excess, then numerous pores were visible in the composite powder, and the combination of particles was poor.When the amount of Mo–B_4_C was 35.2%, the mechanical properties of the prepared coating reached optimal values: minimum porosity of 0.31 ± 0.15%, bonding strength of 77.81 ± 1.77 Mpa, nanoindentation hardness of 20.12 ± 1.85 GPa, Young’s modulus of 281.52 ± 30.22 GPa, and fracture toughness of 6.38 ± 0.45 MPa·m^1/2^.The strengthening mechanism of Mo–B_4_C in WC–Co composite coatings was as follows: the addition of Mo–B_4_C reacted with Co to form ternary borides CoMo_2_B_2_ and CoMoB. Owing to the in-situ reaction, a perfect interface formed between Co and ternary boride. Simultaneously, ternary boride also formed a perfect interface with Mo and WC of a particular crystallographic orientation. Therefore, the addition of Mo–B_4_C improved the interface with WC–Co, forming a dense microstructure. This change improved the mechanical properties and corrosion resistance property of the coating.

## Figures and Tables

**Figure 1 materials-13-01995-f001:**
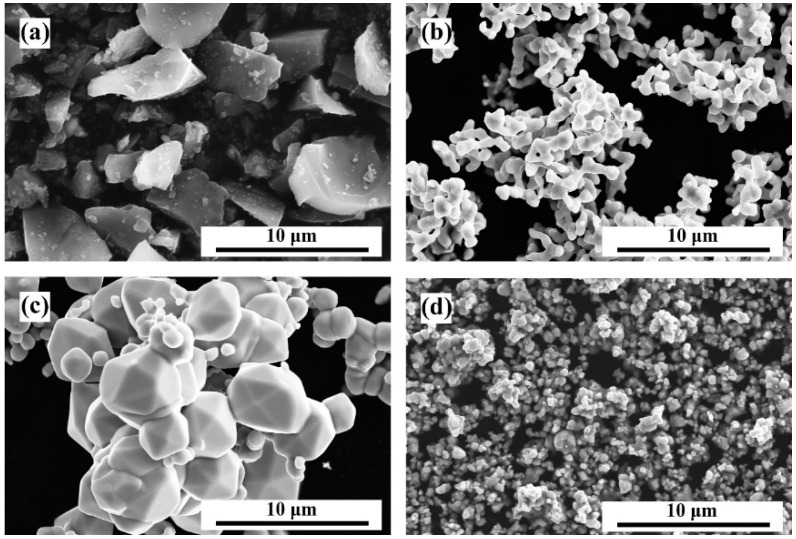
Scanning electron microscopy (SEM) micrographs of raw material powders: (**a**) B_4_C; (**b**) Co; (**c**) Mo; (**d**) WC.

**Figure 2 materials-13-01995-f002:**
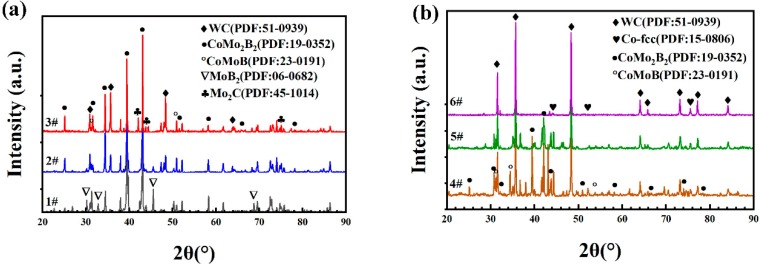
X-ray diffraction (XRD) patterns of composite powders with different raw material compositions: (**a**) 1#, 2#, 3#; (**b**) 4#, 5#, 6#.

**Figure 3 materials-13-01995-f003:**
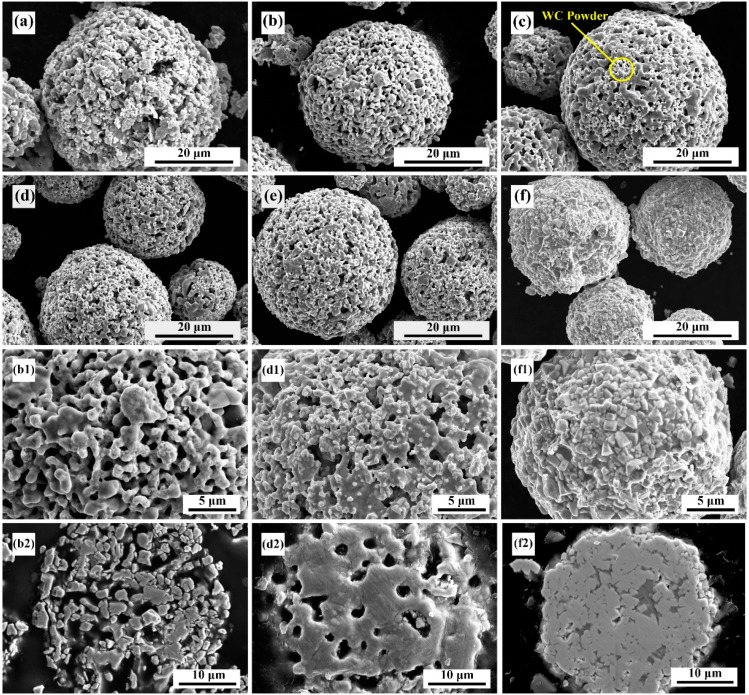
SEM micrographs of cross-sectional microstructures of composite powders: (**a**) #1; (**b**) #2; (**c**) #3; (**d**) #4; (**e**) #5; (**f**) #6. High-magnification micrographs of composite powders: (**b1**) #2; (**d1**) #4; (**f1**) #6. Micrographs of cross-sectional microstructures of composite powders: (**b2**) #2; (**d2**) #4; (**f2**) #6.

**Figure 4 materials-13-01995-f004:**
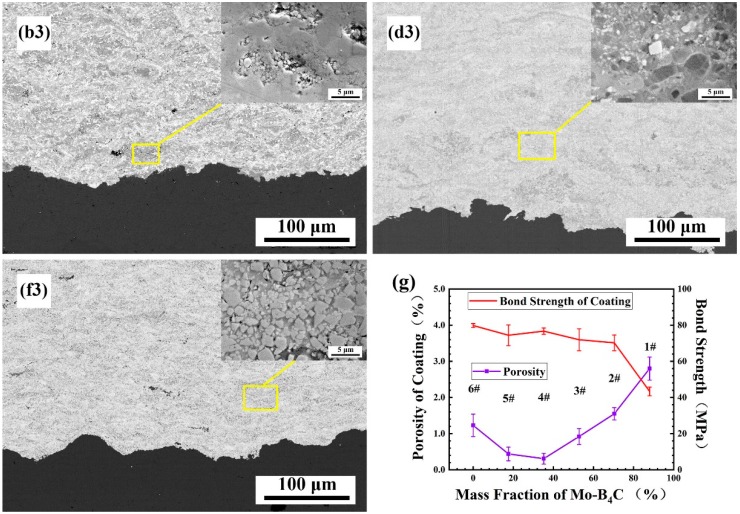
Cross-sectional microstructure of coating with different Mo–B_4_C additions: (**b3**) #2; (**d3**) #4; (**f3**) #6. (**g**) Porosities and bond strengths of the coatings.

**Figure 5 materials-13-01995-f005:**
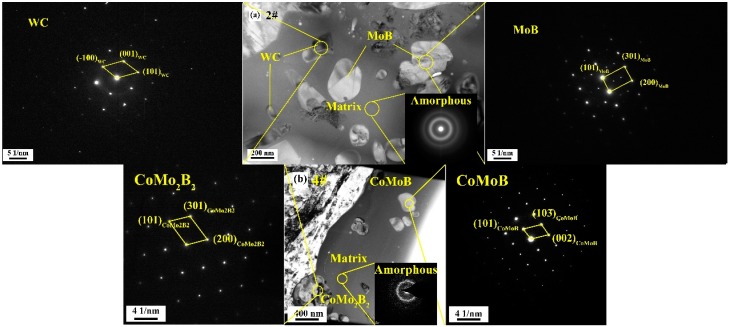
**Transmission Electron Microscope** (TEM)s and selected area electron diffraction diagrams of coatings: (**a**) #2; (**b**) #4.

**Figure 6 materials-13-01995-f006:**
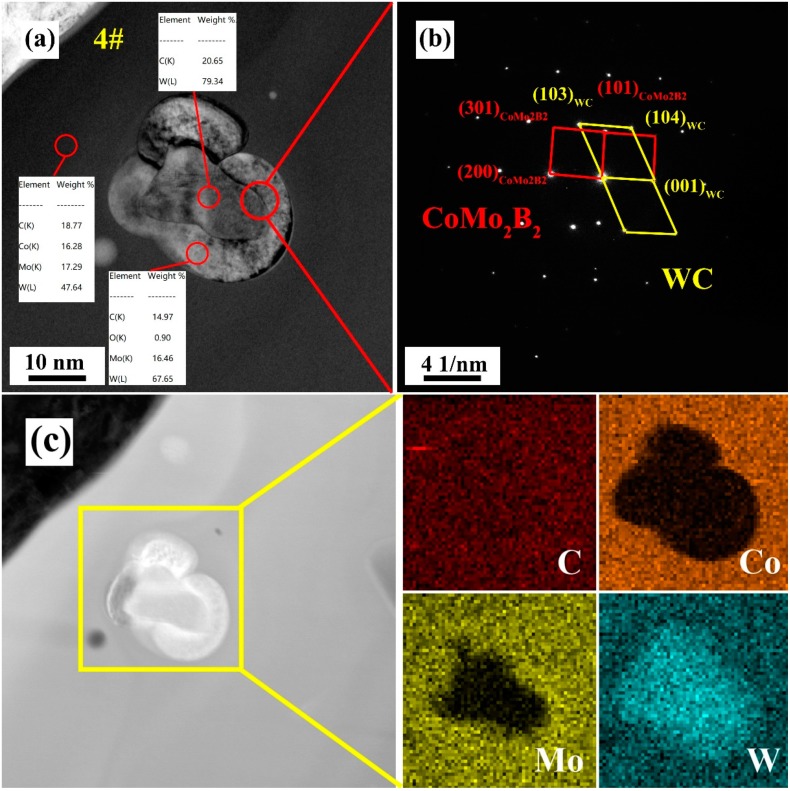
(**a**) Bright field image and EDS of #4 coating; (**b**) selected area electron diffraction diagram of 4# coating; (**c**) high-angle annular dark-field scanning transmission electron microscopy (HAADF-STEM) mapping of #4 coating.

**Figure 7 materials-13-01995-f007:**
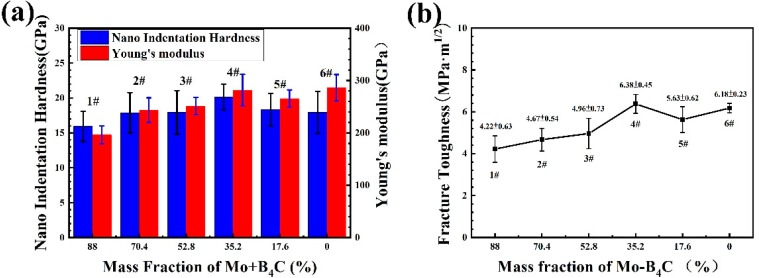
(**a**) Nanoindentation hardnesses and Young’s moduli of coatings; (**b**) fracture toughnesses of coatings.

**Figure 8 materials-13-01995-f008:**
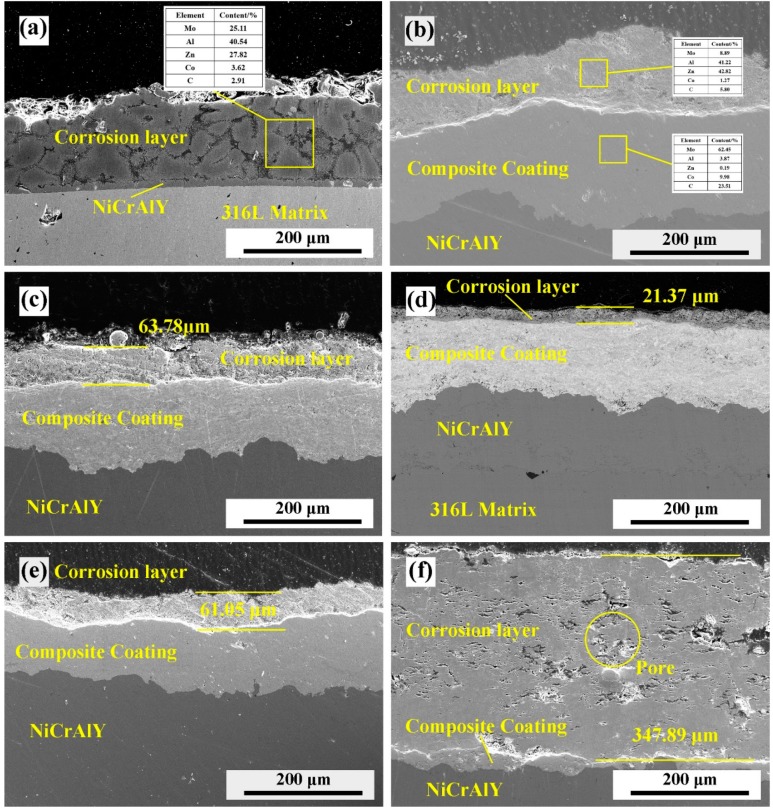
SEM micrographs of cross-sectional microstructures of composite coating after 72 h and 630 °C molten zinc aluminum corrosion: (**a**) #1; (**b**) #2; (**c**) #3; (**d**) #4; (**e**) #5; (**f**) #6.

**Table 1 materials-13-01995-t001:** Raw material composition of composite powders (wt. %).

Sample	WC	Mo, B_4_C			Co
		Mo + B_4_C	Mo	B_4_C	
1#	0	88.0	78.8	9.2	12.0
2#	17.6	70.4	63.1	7.3	12.0
3#	35.2	52.8	47.3	5.5	12.0
4#	52.8	35.2	31.5	3.7	12.0
5#	70.4	17.6	15.8	1.8	12.0
6#	88.0	0	0	0	12.0

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
