# Peer review of "In Situ Ternary Boride: Effects on Densification Process and Mechanical Properties of WC-Co Composite Coating"

_materials, 2020, doi:10.3390/ma13081995_

Round 1
Reviewer 1 Report
The manuscript entitled, “Effect of ternary boride formed in situ by Mo-B4C and WC-Co on densification process and mechanical properties of composite coating” by Bao et al. reports the development of protective coating for sink rolls used in hot dip galvanizing process. They have used SEM and HR-TEM to characterize the microstructure of raw materials and coatings and studied the mechanical properties of the coatings. The microstructure characterization data is clearly support the mechanical properties of coatings. Although the authors report very nice properties and characterization data which suite the manuscript for publication in Materials, the major concerns are structure and presentation of results, as well as writing. The current version is somewhat confusing and needs extensive editing and rephrasing in all section.
Author Response
Dear Reviewers:
Thank you for your letter and the reviewers’ comments on our manuscript entitled “Effect of ternary boride formed in situ by Mo-B4C and WC-Co on densification process and mechanical properties of composite coating” (No.: materials-751777). We greatly appreciate the editor and reviewers for their significant investment of time in reviewing our manuscript. The corresponding corrections have been made and the comments have been addressed carefully. We hope these revisions have satisfied the reviewers. The main corrections in the paper and the responses to the reviewer’s comments are as follows:
To Reviewer 1:
Comments: The manuscript entitled, “Effect of ternary boride formed in situ by Mo-B4C and WC-Co on densification process and mechanical properties of composite coating” by Bao et al. reports the development of protective coating for sink rolls used in hot dip galvanizing process. They have used SEM and HR-TEM to characterize the microstructure of raw materials and coatings and studied the mechanical properties of the coatings. The microstructure characterization data is clearly support the mechanical properties of coatings. Although the authors report very nice properties and characterization data which suite the manuscript for publication in Materials, the major concerns are structure and presentation of results, as well as writing. The current version is somewhat confusing and needs extensive editing and rephrasing in all section.
Reply: We sincerely appreciate the thorough and attentive review. In order to show our research results more clearly, we have made detailed modifications to the whole article to ensure that it is more logical. And we revised the language of the full article in order to express our research more accurately. The new addition of the article is highlighted in yellow.

Reviewer 2 Report
This paper entitled “Effect of ternary boride formed in situ by Mo-B4C and WC-Co on densification process and mechanical properties of composite coating", the authors study and characterize a novel power coating based on Mo-B4C and combined with WC-12Co The topic is of interest to readers of Materials to know about the new insight of coating technologies and its influence in dynamic loading and microstructure changes the enhance of mechanical performance and its microstructure design. However, the manuscript presents a poor discussion and there are a lot of English grammar mistakes and not common logical construction of sentences, so the action to consider this paper for publication are the following one:
- The introduction section needs to be improved because it is vague and not well describe the contribution until the date and what is bringing new this research. It is recommended to explain individually each contribution and not merge all of them in one single reference. Also, there is not a rational part at the end of the introduction explaining which one is the novelty of this work respect to previous works on the same topic.
- Please a thoroughly English revision of the manuscript is needed to improve the cohesion and minimize the grammar mistakes.
- A poor discussion has been done with no references at all to compare the results with previous experiments. So it cannot be determined which is the contribution of this research compare the results with previous authors.
Author Response
Dear Reviewers:
Thank you for your letter and the reviewers’ comments on our manuscript entitled “Effect of ternary boride formed in situ by Mo-B4C and WC-Co on densification process and mechanical properties of composite coating” (No.: materials-751777). We greatly appreciate the editor and reviewers for their significant investment of time in reviewing our manuscript. The corresponding corrections have been made and the comments have been addressed carefully. We hope these revisions have satisfied the reviewers. The main corrections in the paper and the responses to the reviewer’s comments are as follows:
To Reviewer 2:
Comments: This paper entitled “Effect of ternary boride formed in situ by Mo-B4C and WC-Co on densification process and mechanical properties of composite coating", the authors study and characterize a novel power coating based on Mo-B4C and combined with WC-12Co The topic is of interest to readers of Materials to know about the new insight of coating technologies and its influence in dynamic loading and microstructure changes the enhance of mechanical performance and its microstructure design. However, the manuscript presents a poor discussion and there are a lot of English grammar mistakes and not common logical construction of sentences, so the action to consider this paper for publication are the following one.
Response: We sincerely appreciate the thorough and attentive review. We have carefully revised the manuscript based on the comments.
Comment 1: The introduction section needs to be improved because it is vague and not well describe the contribution until the date and what is bringing new this research. It is recommended to explain individually each contribution and not merge all of them in one single reference. Also, there is not a rational part at the end of the introduction explaining which one is the novelty of this work respect to previous works on the same topic.
Response: We have revised the logic and expression of the introduction, hoping that we can more clearly express the basis and significance of our research. At the same time, we added some content at the end to highlight the significance of our research, which highlighted in yellow:
“A low-porosity coating rich in CoMoB and CoMo2B2 ternary boride, which could enhance the resistance to melting zinc corrosion, was successfully prepared with mechanical properties equivalent to those of traditional WC–Co coatings, but stronger resistance to molten zinc corrosion due to the ternary boride.”
Comment 2: Please a thoroughly English revision of the manuscript is needed to improve the cohesion and minimize the grammar mistakes.
Response: We sincerely appreciate the thorough and attentive review. In order to show our research results more clearly, we have made detailed modifications to the whole article to ensure that it is more logical. And we revised the language of the full article in order to express our research more accurately. The new addition of the article is highlighted in yellow.
Comment 3: A poor discussion has been done with no references at all to compare the results with previous experiments. So, it cannot be determined which is the contribution of this research compare the results with previous authors.
Response: Thanks for your kindly suggestion. We have tried to add references in the discussion part, especially in the part of B4C improving the interface between WC and Co. We added the research results we draw on. At the end of the paper, we added the performance comparison of the composites coatings prepared, WC-Co and pure ceramic to reveal our advantage is that we take into account the toughness and resistance to molten zinc corrosion.

Reviewer 3 Report
In this work, different proportions of Mo-B4C powder have been added on WC-12Co, and the composite powder has been prepared via mixing, granulation and heat treatment. Then a corrosion-resistant coating of high-temperature zinc solution has been prepared by supersonic flame spraying. Various characterization methods such as XRD, TEM, SEM, HADDF-STEM, EDS, and mechanical analysis have been employed to study the effect of Mo-B4C on the microstructure, phase, porosity, bonding strength and elastic modulus of the composite powder and coating. The work seems interesting and it can be accepted after minor revision:
- The manuscript is not in the right template of the journal. The reference style is not also in the right style requested by Materials.
- Some sentences like the first sentence in the abstract are too long and less informative. Therefore, it is highly suggested shortening the sentences as much as possible to be easy to follow them.
- The majority of the references are rather old and it is recommended to replace them with the new works.
- Have the outcomes of this work been comparable with the coatings which are present in the literature?
Author Response
Dear Reviewers:
Thank you for your letter and the reviewers’ comments on our manuscript entitled “Effect of ternary boride formed in situ by Mo-B4C and WC-Co on densification process and mechanical properties of composite coating” (No.: materials-751777). We greatly appreciate the editor and reviewers for their significant investment of time in reviewing our manuscript. The corresponding corrections have been made and the comments have been addressed carefully. We hope these revisions have satisfied the reviewers. The main corrections in the paper and the responses to the reviewer’s comments are as follows:
To Reviewer 3:
Comments: In this work, different proportions of Mo-B4C powder have been added on WC-12Co, and the composite powder has been prepared via mixing, granulation and heat treatment. Then a corrosion-resistant coating of high-temperature zinc solution has been prepared by supersonic flame spraying. Various characterization methods such as XRD, TEM, SEM, HADDF-STEM, EDS, and mechanical analysis have been employed to study the effect of Mo-B4C on the microstructure, phase, porosity, bonding strength and elastic modulus of the composite powder and coating. The work seems interesting and it can be accepted after minor revision:
Reply: We sincerely appreciate the thorough and attentive review. We have carefully revised the manuscript based on the comments.
Comment 1: The manuscript is not in the right template of the journal. The reference style is not also in the right style requested by Materials.
Response: Thanks for your kindly suggestion. We downloaded the paper template from the official website of ‘materials’, and have revised the paper, hoping to meet the requirements of the journal.
Comment 2: Some sentences like the first sentence in the abstract are too long and less informative. Therefore, it is highly suggested shortening the sentences as much as possible to be easy to follow them.
Response: We have carefully revised our paper according to your suggestion, and try to reduce the long sentence. At the same time, we modify the language and logic, hoping to express our research content more accurately.
Comment 3: The majority of the references are rather old and it is recommended to replace them with the new works.
Response: I'm sorry that we can't find more new references. Because the resistant to molten zinc corrosion coating is a relatively old technology, we have encountered some problems in the actual production, such as its short life, while the nano ceramic coating will greatly increase the cost. After investigating some relatively references, we learned that in-situ reaction can solve the interface problem, so we applied this idea to the preparation of new composite coatings, and achieved some results. We would like to share our research with others and solve the problems in the actual production.
Comment 4: Have the outcomes of this work been comparable with the coatings which are present in the literature?
Response: At the end of the paper, we added the performance comparison of the composite coatings prepared, WC-Co and pure ceramic to reveal our advantage is that we take into account the toughness and resistance to molten zinc corrosion. The addition parts have been highlighted in yellow.
“The mechanical properties of the in-situ ternary boride composite coating are comparable to that of WC Co, and the ternary boride can effectively prevent the coating from being corroded by high temperature molten zinc. The corrosion-resistant wc-wb-w2b ceramics prepared by Sugiyama et al [18], have an initial fracture toughness of less than 5.8 MPa·m1/2 and a young's modulus of up to 700 GPa. In contrast, the ternary boride composite coating has the advantages of mechanical properties, and the difference of Young's modulus between the ternary boride composite coating and the matrix is smaller. Compared with ceramic coating, the composite coating prepared has better toughness and adaptability.”

Reviewer 4 Report
The manuscript present several experimental results regarding the preparation of a composite powder based on addition of Mo-B4C to WC-Co ones in various ratios, which further is applied on stainless steel substrates using supersonic flame spraying and an intermediary bond layer of NiCrAlY alloy. Both composite powders and the coating layers have been morphologically and structurally characterized involving SEM, XRD, TEM, HAADF-STEM and from composition view point using EDS analysis. In addition the mechanical characteristics of the formed protective coatings have been investigated.
The paper presents some interesting results.
However the entire manuscript is recommended to be re-written. It cannot be published in the present form. It appears rather as a technical report, not a scientific paper. Moreover, the entire contribution has to be drastically checked from English grammar and phraseology viewpoints.
Only some recommendations are given below because otherwise the reviewer should have written almost the entire paper again.
Abstract: please reformulate the entire abstract:
“The paper presents experimental results regarding the synthesis and characterization of several composite powders based on addition of Mo-B4C to WC-Co ones in various ratios able to be then applied onto stainless steel substrate as protective coatings by supersonic flame spraying. Both the prepared composite powders and the coating layers have been thoroughly investigated involving a large range of techniques including SEM, TEM, HADDF-STEM microscopies, X-ray diffraction and EDS. Moreover, the influence of the Mo-B4C addition on the mechanical properties of the obtained coatings have been assessed considering the microstructure, phase, porosity, bonding strength and elastic modulus of the prepared composite powders . The results showed that the use of an appropriate amount of Mo-B4C may facilitate the reaction with Co forming ternary CoMo2B2 and CoMoB borides which built a perfect continuous interface that contributed to the improvement of the mechanical characteristics of the coating. The optimum amount of Mo-B4C to be added was found to be 35.2%, which then allowed improved mechanical properties, respectively: a minimum porosity of 0.31±0.15%, the coating bonding strength of about 77.81± 1.77 MPa, the nano-indentation hardness of 20.12±1.85 GPa, the Young's modulus of 281.52±30.22 GPa and the fracture toughness of 6.38±0.45 MPa·m1/2.”
The authors didn’t report any corrosion related investigations. Therefore it is recommended to avoid these corrosion issues. Otherwise, a minimum corrosion performance investigation to prove the improvement should be added.
Introduction section:
Lines 24-25, please correct: “ Hot dip galvanizing represents a cost effective and efficient process to prevent steel corrosion…..”
Lines 31-32: please reformulate- the sentence is unclear.
Please correct: …”under premises…” all over the manuscript.
Line 45: please correct: “In recent years….”
Line 48, please reformulate: “..an increased interest has been noticed on the study of….”
Line 68: please correct: ….” under which the steel is softened due to high temperature…”
Line 74: please correct: …..”were investigated”
Lines 77-78: please replace “improve of corrosion resistane” because no investigations are reported.
Line 79: please correct: “2. Matherials and methods”. In addition please number the corresponding subsections (i.e. 2.1- Preparation of composite powder; 2.2 –Preparation of the coatings, etc.)
Lines 81-87: please reformulate: “WC powder (Xiamen Jinlu Special Alloy Co., LTD, average particle size of 1.10 µm), Mo powder (BGRIMM Advanced Material Science & Technology Co., LTD, average particles size of 1.75 µm), Co powder (Zhejiang Huayou Cobalt Industry Co., LTD, average particles size of 1.52 µm) and B4C powder (Nanuo Advanced Materials Co., LTD, average particles size of 1.95 µm) were selected as raw materials.
Lines 88-101: please reformulate the entire section: “ The specific process of powders mixing consists in the following steps: (i) mix of the 4 types of powders according to the compositions presented in Table 1; (ii) selection of the planetary ball mill for full blending….(iii) addition of 3 wt % polyethylene glycol (PEG4000), using deionized water as mill additive. The full mixing process was performed for 24 h, at a selected speed of 80 rad/min. During the granulation process a LGZ-50 centrifugal spray dryer has been used……….” Etc.
Lines 104-105: please reformulate: …” stainless steel was used as metallic substrate”.
Lines 112-113: please reformulate: The influence of Mo-B4C addition on the microstructure, morphology, densification and mechanical 1 properties of the composite powder and of the coating has been investigated in detail.”
Lines 132-133: please correct: …”fracture toughness of the coating was calculated using equation (1).
Please use term “equation” or “relationship” instead of “formula” (i.e. lines 162-163)
Line 139: please reformulate: “3.1. Characterization of the raw materials”
Fig. 1: the morphology of WC (Fig.1d) is really irregular? Please check.
Line 148: please reformulate: 3.2. Characterization of the prepared composite powders
Figure 2 should be redrawn in classical way to better observe the pattern of the powders and the number of the specimens. What cards have been used to identify the phases? Please mention.
The XRD discussion should be more detailed.
Please add references to sustain you reaction mechanism according to equations 2 and 3.
Lines 175-177: The authors declared the formation of ternary borides but no evidence of this was provided. Please detail.
Line 277: please reformulate: 3.5. The influence of -B4C addition on the mechanical properties of the coatings

Author Response
Dear Reviewers:
Thank you for your letter and the reviewers’ comments on our manuscript entitled “Effect of ternary boride formed in situ by Mo-B4C and WC-Co on densification process and mechanical properties of composite coating” (No.: materials-751777). We greatly appreciate the editor and reviewers for their significant investment of time in reviewing our manuscript. The corresponding corrections have been made and the comments have been addressed carefully. We hope these revisions have satisfied the reviewer.
We sincerely appreciate the thorough and attentive review. In order to show our research results more clearly, we have made detailed modifications to the whole article to ensure that it is more logical. And we revised the language of the full article in order to express our research more accurately. The new addition of the article is highlighted in yellow.
I need to reply to you with some details:
1.We revised the abstract and title of this article.
- In the whole article, we can't show the experiment about the molten zinc corrosion resistance of the coating we prepared. I'm sorry that we haven't completely completed this part of the experiment. However, we have made some progress and confirmed that the coating we prepared has better corrosion resistance than WC-Co coating. In the introduction, we quote the previous research, which shows that boride coating has a good effect on molten zinc corrosion resistance, which is also the original intention of our preparation of composite coating. If delete this part, the meaning of the coating we prepared could not be understood.

Round 2
Reviewer 1 Report
The new version of manuscript has been improved. However, there are some minor comments need to be addressed as stated below.
(1) Page#3, line#118: The sentence, "....... the coating thickness was 250 mm" seems to be wrong. If so, please correct it.
(2) Page#8, line#247: The sentence, "coating #4 showed the highest bonding strength, owing to the increase in bonding strength and reduction in defects" does not make sense.
(3) Page#8, line#265: the state, " lowest densification" need to be changed to lowest porosity or highest densification.
Author Response
Dear Reviewers:
Thank you for your letter and the reviewers’ comments on our manuscript entitled “Effect of ternary boride formed in situ by Mo-B4C and WC-Co on densification process and mechanical properties of composite coating” (No.: materials-751777). We greatly appreciate the editor and reviewers for their significant investment of time in reviewing our manuscript. The corresponding corrections have been made and the comments have been addressed carefully. We hope these revisions have satisfied the reviewers. The main corrections in the paper and the responses to the reviewer’s comments are as follows:
To Reviewer 1:
Comments1: Page#3, line#118: The sentence, "....... the coating thickness was 250 mm" seems to be wrong. If so, please correct it.
Reply: Thank you for your kindly suggestion, and we have revised the wrong unit “mm” to “μm”. And the corrections line #118 has been highlighted yellow.
Comments2: Page#8, line#247: The sentence, "coating #4 showed the highest bonding strength, owing to the increase in bonding strength and reduction in defects" does not make sense.
Reply: Thank you for your kindly suggestion, and we have revised the manuscript. This sentence has been revised to “coating #4 showed the highest bonding strength, owing to the reduction in defects.”.
Comments3: Page#8, line#265: the state, " lowest densification" need to be changed to lowest porosity or highest densification.
Reply: We have changed the "lowest densification" to lowest porosity.

Reviewer 2 Report
Despite that the English grammar had been greatly improved, the aim of the paper had not fully assessed. The main objective is to investigate the corrosion of this material, but any test of corrosion is performed. The research work present a thorough analysis of material characerization, but it is not investigated is performance in salt chamber test. In my opinion, this paper can not be accepted until this corrosion tests validate that this materials presents good corrosion properties. So I recommend to reject and resubmit.
Author Response
Dear Reviewers:
Thank you for your letter and the reviewers’ comments on our manuscript entitled “Effect of ternary boride formed in situ by Mo-B4C and WC-Co on densification process and mechanical properties of composite coating” (No.: materials-751777). We greatly appreciate the editor and reviewers for their significant investment of time in reviewing our manuscript. The corresponding corrections have been made and the comments have been addressed carefully. We hope these revisions have satisfied the reviewers. The main corrections in the paper and the responses to the reviewer’s comments are as follows:
To Reviewer 2:
Comments: Despite that the English grammar had been greatly improved, the aim of the paper had not fully assessed. The main objective is to investigate the corrosion of this material, but any test of corrosion is performed. The research work present a thorough analysis of material characerization, but it is not investigated is performance in salt chamber test. In my opinion, this paper can not be accepted until this corrosion tests validate that this materials presents good corrosion properties. So I recommend to reject and resubmit.
Reply: Thank you for your kindly suggestion, and we have revised the manuscript. We have added the performance in molten zinc aluminum corrosion. And the added parts were highlighted in yellow.

Reviewer 4 Report
The authors made partially the recommended corrections and a partial improvement of the manuscript has been observed.
However, other aspects still need to be considered before publication.
Again: I understand the experiments related to corrosion behavior are not finalized. Consequently I have to mention again: please don’t refer to corrosion issues in this manuscript as far as you don’t discuss this aspect.
You cannot start the Abstract referring to corrosion while the manuscript doesn’t provide any information onthis issue. Please reformulate and please take into considerations my previous recommendations.
For a better understanding and logical structuring of the information, I come back again with the following recommendations:
Please correct: “2. Materials and methods”.
please reformulate: “3.1. Characterization of the raw materials”
please reformulate: 3.2. Characterization of the prepared composite powders
Figure 2 should be redrawn in classical way to better observe the pattern of the powders and the number of the specimens. What cards have been used to identify the phases? Please mention.

Author Response
Dear Reviewers:
Thank you for your letter and the reviewers’ comments on our manuscript entitled “Effect of ternary boride formed in situ by Mo-B4C and WC-Co on densification process and mechanical properties of composite coating” (No.: materials-751777). We greatly appreciate the editor and reviewers for their significant investment of time in reviewing our manuscript. The corresponding corrections have been made and the comments have been addressed carefully. We hope these revisions have satisfied the reviewers. The main corrections in the paper and the responses to the reviewer’s comments are as follows:
To Reviewer 4:
Comments1: The authors made partially the recommended corrections and a partial improvement of the manuscript has been observed.
However, other aspects still need to be considered before publication.
Again: I understand the experiments related to corrosion behavior are not finalized. Consequently I have to mention again: please don’t refer to corrosion issues in this manuscript as far as you don’t discuss this aspect.
You cannot start the Abstract referring to corrosion while the manuscript doesn’t provide any information onthis issue. Please reformulate and please take into considerations my previous recommendations.
Reply: Thank you for your kindly suggestion, and we have revised the manuscript. Due to the main objective to investigate the corrosion of this material, we have added the performance in molten zinc aluminum corrosion. And the added parts were highlighted in yellow.
Comments2: For a better understanding and logical structuring of the information, I come back again with the following recommendations:
Please correct: “2. Materials and methods”.
please reformulate: “3.1. Characterization of the raw materials”
please reformulate: 3.2. Characterization of the prepared composite powders
Reply: Thank you for your kindly suggestion, and we have revised the manuscript, which were highlighted in yellow.
Comments3: Figure 2 should be redrawn in classical way to better observe the pattern of the powders and the number of the specimens. What cards have been used to identify the phases? Please mention.
Reply: Thank you for your suggestion, and we have redrawn the Fogure.2 in classical way, and provided the pdf-card number in Figure.2. We hope these revisions have satisfied the reviewers.
